# The p.Pro482Ala Variant in the CNNM2 Gene Causes Severe Hypomagnesemia Amenable to Treatment with Spironolactone

**DOI:** 10.3390/ijms23137284

**Published:** 2022-06-30

**Authors:** Ioannis Petrakis, Eleni Drosataki, Ioanna Stavrakaki, Kleio Dermitzaki, Dimitra Lygerou, Myrto Konidaki, Christos Pleros, Nikolaos Kroustalakis, Sevasti Maragkou, Ariadni Androvitsanea, Ioannis Stylianou, Ioannis Zaganas, Kostas Stylianou

**Affiliations:** 1Nephrology Department, Heraklion University Hospital, 71500 Heraklion, Greece; petrakgia@gmail.com (I.P.); elenidro2@hotmail.com (E.D.); giannastavrakaki@hotmail.gr (I.S.); ekderm@gmail.com (K.D.); dimitra.ligerou@gmail.com (D.L.); myrtokonidaki@gmail.com (M.K.); xpleros@gmail.com (C.P.); nkroustalakis@yahoo.gr (N.K.); sevastimaragou@gmail.com (S.M.); ariaandrovitsanea@gmail.com (A.A.); 2Institute of Applied and Computational Mathematics of FORTH (Foundation of Research and Technology-Hellas), 70013 Heraklion, Greece; steelgiannis@gmail.com; 3Neurogenetics Laboratory Medical School, University of Crete, 71003 Heraklion, Greece; johnzag@yahoo.com

**Keywords:** hypomagnesemia, HOMG6, CNNM2, whole exome sequencing

## Abstract

Renal hypomagnesemia syndromes involving *CNNM2* protein pathogenic variants are associated with variable degrees of neurocognitive dysfunction and hypomagnesemia. Here, we report a family with a novel *CNNM2* p.Pro482Ala variant, presenting with overt hypomagnesemia and mild neurological involvement (autosomal dominant renal hypomagnesemia 6, HOMG6, MIM# 613882). Using a bioinformatics approach, we showed that the p.Pro482Ala amino acid substitution causes a 3D conformational change in *CNNM2* structure in the cystathionin beta synthase (CBS) domain and the carboxy-terminal protein segment. A novel finding was that aldosterone inhibition with spironolactone helped to alleviate hypomagnesemia and symptoms in the proband.

## 1. Introduction

Renal magnesium handling involves various ion transporters within the nephron [1]. M2 cyclin and CBS domain divalent metal cation transport mediator 2 (CNNM2) belongs to the major magnesium-handling proteins of the kidney [2,3]. *CNNM2* mRNA is abundantly present within the kidneys, brain, and parathyroid glands [4].

Localization experiments as well as data from *Cnnm2* knock-out mice confirmed that CNNM2 is localized in the basolateral membrane of distal tubular cells [2,5]. More specifically, the incorporation of the CNNM2 protein in the cell membrane of HEK293 cells is associated with increased magnesium uptake. Knocking down *Cnnm2* gene in mice results in renal magnesium wasting and abnormal neuronal development [1,2]. *CNNM2* gene polymorphisms have been associated with increased risk for schizophrenia [6], hypertension, and coronary artery disease in specific ethnic groups [7].

Beyond hypomagnesemia, *CNNM2* variants display various clinical manifestations from the central nervous system (CNS) ranging from headache to epileptic seizures, mental retardation [3], and structural malformations of the brain tissue [8,9]. Based on the severity of the neurological manifestations, *CNNM2* pathogenic variants may manifest either as a mere hypomagnesemia syndrome with mild neurological manifestations (autosomal dominant renal hypomagnesemia 6, HOMG6, MIM# 613882) or a syndrome of hypomagnesemia, seizures, and mental retardation (HSMR1 MIM# 616418), frequently associated with obesity and autism.

Herein, we present a novel *CNNM2* variant associated with severe hypomagnesemia and mild neurological phenotype (HOMG6) in a Greek family and discuss our therapeutic approach with spironolactone to restore normal serum magnesium levels. Furthermore, we construct a three-dimensional (3D) model of the mutated protein and compare it with the native one.

### Patients and Methods

Informed consent was obtained from all family members. Whole Exome Sequencing (WES) was utilized for the identification of the responsible mutation in the index patient, as previously described [10,11]. Sanger sequencing analysis was used to verify or exclude the presence of the variant in all available family members. Laboratory values and clinical characteristics were obtained by reviewing patients’ clinical records. The computational assessment of the pathogenicity of the mutation was carried out by using the Ingenuity Variant Analysis Bioinformatics Software (Qiagen) and Varsome software [12]. Given a known amino acid sequence, the 3D protein structure was predicted using the neural network database AlphaFold-2 (EMBL-EBI). AlphaFold-2 achieved a high accuracy level, comparable to standard experimental techniques such as X-ray Crystallography [13]. The prediction was then loaded as a .pdb file in the molecular viewer PyMOL, in order to view and prepare the figures. PyMOL Software (Version 2.5.2, Schrödinger) was utilized for protein tertiary structure comparison and prediction.

## 2. Results

The index patient is a 36-year-old Caucasian female presenting with headache, muscle aches, and paresthesias of the upper extremities. The patient’s medical records revealed persistent hypomagnesemia since her first pregnancy (at the age of 25 years), irresponsive to oral magnesium supplements. Other causes of hypomagnesemia such as diuretics, proton pump inhibitors, diarrhea, and diabetes mellitus were excluded. Serum magnesium without oral magnesium supplements averaged a value of 1 mg/dL (0.41 mmol/L), while on supplements it averaged 1.5 mg/dL associated with increased fractional excretion of magnesium in the urine (FeMg^2+^ ranging from 10% to 20%). All other laboratory values including renal function and parathormone were within normal limits. A brain magnetic resonance imaging (MRI) showed two isolated demyelinating-type lesions near the corpus callosum, raising the possibility of familial renal magnesium wasting. Renal hypomagnesemia was also verified in her son (FeMg^2+^ ranging from 10% to 17%) but not in her daughter and sister. The son suffered from attention deficit and hyperactivity disorder. At the time of examination, his hypomagnesemia could be alleviated with oral magnesium supplementation (serum magnesium 1.6–1.8 mg/dL). None of the family members exhibited obesity, seizures, or intellectual disability.

WES revealed a novel heterozygous mutation in the *CNNM2* gene (c.1444C>G, p.Pro482Ala) that was classified as likely pathogenic in computational assessment by the Varsome [12], according to the American College of Medical Genetics (ACMG) criteria [14]: PM2, PM6, PP3, and PP2. This variant does not have a gnomAD genomes entry and is similarly absent from other databases such as Allele Frequency Community (AFC) and 1000 genomes. Importantly, it was not found in any of the 201 participants of a local cohort from Crete, from which we have WES data available [15]. The p.Pro482Ala *CNNM2* variant is predicted to cause loss of protein function based on being on a strongly conserved region (PhyloP100way 7.81 higher than cut-off limit of 7.2) of a dominant loss of function gene. Furthermore, the Combined Annotation Dependent Depletion (CADD) score was high (25.8), classifying this variant as deleterious. Finally, all available family members were genotyped by Sanger sequencing and the alteration was found to segregate with the disease in the proband and her son (Figure 1).

Oral administration of large doses of magnesium supplements (8 tablets of 60.77 mg elemental magnesium per day) failed to completely correct hypomagnesemia and symptoms in the proband. Thus, we opted to add the magnesium-sparing diuretic spironolactone at a dose of 25 mg per day, in an effort to increase distal tubular magnesium reabsorption. This maneuver resulted in restoration of serum magnesium levels up to 1.7–1.9 mg/dL with fewer magnesium pills (from eight to four tablets per day).

The p.Pro482Ala amino acid substitution was found to affect the 3D structure of the CBS domain and the carboxy-terminal region of the CNNM2 protein (Figure 2 and Figure 3).

## 3. Discussion

The CNNM2 pathogenic variants described so far appear with a wide spectrum of severe neurological manifestations [16]. The importance of magnesium in maintaining neuronal synapses and signaling is pivotal, not only in experimental models but also in humans [17]. Magnesium is necessary for proper myelination and exposure to a hypomagnesemic environment can result in irreversible neurological damage in early stages of neural development [18]. It is possible that amino acid replacements in CBS region cause protein conformational changes that affect the binding site of Mg^2+^-ATP complexes, which is responsible for Mg^2+^ extrusion from tubular cells to the circulation [19,20,21].

In the present report, we describe the clinical manifestations of a novel variant in the *CNNM2* gene, causing renal hypomagnesemia type 6. Although this change leads to severe symptomatic hypomagnesemia, it is accompanied by a mild CNS involvement. Using a bioinformatics approach, we showed that the p.Pro482Ala substitution causes a 3D conformational change in the CBS and the carboxy-terminal domain of the CNNM2 protein.

Spironolactone can reduce magnesium loss in the urine in various clinical settings [22,23,24]. It has been proposed that inhibition of sodium permeability by spironolactone in the late DCT hyperpolarizes the apical membrane, thereby increasing the driving force for Mg^2+^ entry into the cells of the DCT [24]. In the present case, spironolactone helped to restore serum magnesium levels without the need for intravenous magnesium supplementation and with tolerable oral magnesium load. Thus, spironolactone can be used as a magnesium-sparing agent in this rare disease.

The limitations of the present report include the lack of experimental confirmation, the small number of affected family members, and that spironolactone was tried only in the proband.

In conclusion, the p.Pro482Ala CNNM2 alteration leads to a distinct phenotype characterized by significant hypomagnesemia with mild neurological sequelae, while spironolactone treatment improves the clinical picture.

## Figures and Tables

**Figure 1 ijms-23-07284-f001:**
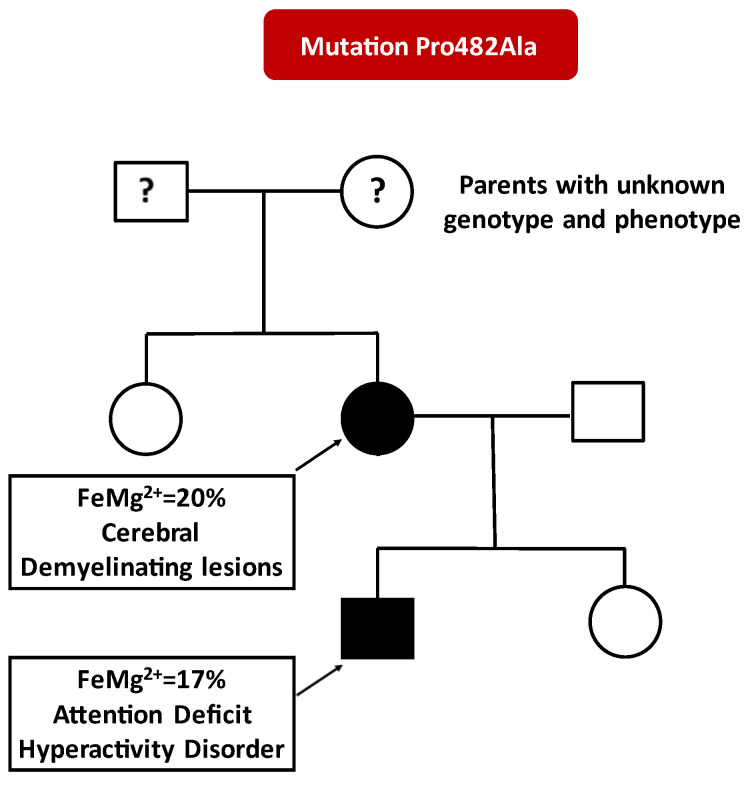
Family tree depicting magnesium fractional excretion (FeMg^2+^) and main clinical characteristics of affected individuals. Black symbols, affected individuals. White symbols, unaffected individuals.

**Figure 2 ijms-23-07284-f002:**
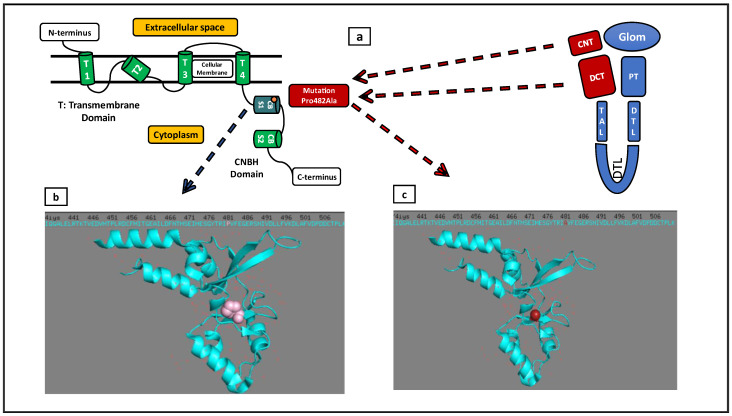
(**a**) Normal localization of CNNM2 (red color) within nephron segments (Glom: Glomerulus, PT: Proximal Tubule, DTL: Descending Thin Limb of Loop of Henle, TAL: Thick Ascending Limb of Loop of Henle, DCT: Distal Convoluted Tubule, CNT: Connecting Tubule). CNNM2 protein consists of four transmembrane domains (T1, T2, T3, T4), two cystathionin beta synthase domains (CBS1, CBS2), and a C-terminal cyclic nucleotide-binding homology (CNBH) domain. The Pro482Ala mutation is localized within the CBS1 domain. (**b**) Representation of Proline (normal variant—pink color) and it’s valency in the CBS1 domain. (**c**) Representation of Alanine (mutant—red color) and its valency in the CBS1 domain.

**Figure 3 ijms-23-07284-f003:**
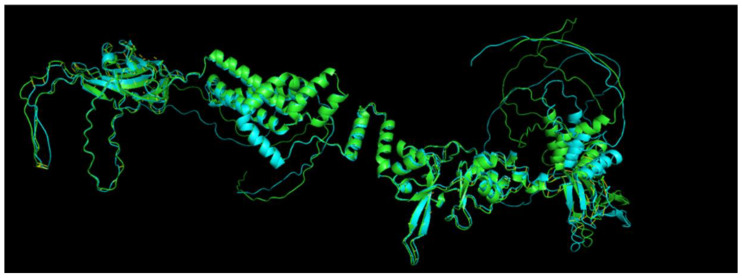
The p.Pro482Ala alteration causes a conformational change in CNNM2 structure in the CBS domain and the carboxy-terminal protein segment, which may affect ATP-Mg^2+^ complex binding to the protein. Mutated protein with blue color and normal protein with green color (protein structure prediction databank Alpha fold-2 [13]).

## Data Availability

The data that support the findings of this study are available from the corresponding author upon reasonable request.

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
