# Peer review of "The p.Pro482Ala Variant in the CNNM2 Gene Causes Severe Hypomagnesemia Amenable to Treatment with Spironolactone"

_ijms, 2022, doi:10.3390/ijms23137284_

Round 1
Reviewer 1 Report
After carreful analysis of the manuscript, I recommend to accept the manuscript: ijms-1770563 “The Pro482Ala mutation in the CNNM2 gene causes severe hypomagnesemia amenable to treatment with spironolactone“.
The manuscript is comprehensive, up to date and includes the relevant and new information.
Selection of International Journal of Molecular Sciences to submit this manuscript is adequate and appropriate.
This manuscript reports that Pro482Ala mutation causes a 3D conformational change in cation transport mediator 2 in CBS region, which leads to a distinct phenotype characterized by significant hypomagnesemia with mild neurological sequelae, while spironolactone treatment improves the clinical picture.
Regarding the originality of this manuscript, the article is novel and interesting.
The proposed type of publication – brief report, is well selected.
The use of the references is appropriate.
The quality of English orthography and grammar in manuscript is adequate.
Author Response
We thank the reviewer for the very positive comments.

Reviewer 2 Report
Dr. Petrakis and colleagues present a study about the discovery of a new mutation (CNNM2 Pro482Ala mutation) in a family affected by renal hypomagnesemia and mild neurological involvement (autosomal dominant renal hypomagnesemia 6, HOMG6, MIM# 613882). By using a bioinformatics approach, the authors found that this mutation provokes a 3D conformational change in the structure of CNNM2. Indeed, they made a trial of therapy by spironolactone that counteract at least partially the hypomagnesemia and symptoms in the proband. The article is generally well written and discussed. I have a concern about two simple things that the authors could make to improve their findings:
1) They could state how much the amino acid that was substituted is phylogenetically conserved in other species and humans
2) They could try to find the same mutations in other unrelated families of the same ancestry to see in how many alleles the mutation is found (classically at least 100 alleles should be genotyped).
Author Response
Reviewer 2
Dr. Petrakis and colleagues present a study about the discovery of a new mutation (CNNM2 Pro482Ala mutation) in a family affected by renal hypomagnesemia and mild neurological involvement (autosomal dominant renal hypomagnesemia 6, HOMG6, MIM# 613882). By using a bioinformatics approach, the authors found that this mutation provokes a 3D conformational change in the structure of CNNM2. Indeed, they made a trial of therapy by spironolactone that counteract at least partially the hypomagnesemia and symptoms in the proband. The article is generally well written and discussed. I have a concern about two simple things that the authors could make to improve their findings:
Response: We thank the reviewer for the positive comments.
1) They could state how much the amino acid that was substituted is phylogenetically conserved in other species and humans
Response: Thank you very much for the suggestion. Indeed, the position is strongly conserved (PhyloP100way=7.8 > 7.2). We have added a relative sentence in the results section as follows: “The p.Pro482Ala CNNM2 variant is predicted to cause loss of protein function based on being on a strongly conserved region (PhyloP100way 7.81 higher than the cut-off limit of 7.2) of a dominant loss of function gene. Furthermore, the Combined Annotation Dependent Depletion (CADD) score, was high (25.8), classifying this variant as deleterious”.
2) They could try to find the same mutations in other unrelated families of the same ancestry to see in how many alleles the mutation is found (classically at least 100 alleles should be genotyped).
Response: We thank the reviewer for the constructive comment. We added a relative sentence in the results section stating: “This variant does not have a gnomAD genomes entry and is similarly absent from other databases such as Allele Frequency Community (AFC) and 1000 genomes. Importantly, it was not found in any of the 201 participants of a local cohort from Crete, from which we have WES data available”.